# Perinatal S-Adenosylmethionine Supplementation Represses *PSEN1* Expression by the Cellular Epigenetic Memory of CpG and Non-CpG Methylation in Adult TgCRD8 Mice

**DOI:** 10.3390/ijms241411675

**Published:** 2023-07-19

**Authors:** Tiziana Raia, Federica Armeli, Rosaria A. Cavallaro, Giampiero Ferraguti, Rita Businaro, Marco Lucarelli, Andrea Fuso

**Affiliations:** 1Department of Experimental Medicine, Sapienza University of Rome, 00161 Rome, Italy; tiziana.raia@uniroma1.it (T.R.); giampiero.ferraguti@uniroma1.it (G.F.); marco.lucarelli@uniroma1.it (M.L.); 2Department of Medico-Surgical Sciences and Biotechnologies, Sapienza University of Rome, 04100 Latina, Italy; federica.armeli@uniroma1.it (F.A.); rita.businaro@uniroma1.it (R.B.); 3Department of Surgery, Sapienza University of Rome, 00161 Rome, Italy; rosaria.cavallaro@uniroma1.it; 4Pasteur Institute, Cenci Bolognetti Foundation, Sapienza University of Rome, 00161 Rome, Italy

**Keywords:** presenilin 1, DNA methylation, non-CpG methylation, MIPs (non-CpG methylation-insensitive primers), Alzheimer’s disease, neurodegeneration, perinatal treatment

## Abstract

DNA methylation, the main epigenetic modification regulating gene expression, plays a role in the pathophysiology of neurodegeneration. Previous evidence indicates that 5′-flanking hypomethylation of *PSEN1*, a gene involved in the amyloidogenic pathway in Alzheimer’s disease (AD), boosts the AD-like phenotype in transgenic TgCRND8 mice. Supplementation with S-adenosylmethionine (SAM), the methyl donor in the DNA methylation reactions, reverts the pathological phenotype. Several studies indicate that epigenetic signatures, driving the shift between normal and diseased aging, can be acquired during the first stages of life, even in utero, and manifest phenotypically later on in life. Therefore, we decided to test whether SAM supplementation during the perinatal period (i.e., supplementing the mothers from mating to weaning) could exert a protective role towards AD-like symptom manifestation. We therefore compared the effect of post-weaning vs. perinatal SAM treatment in TgCRND8 mice by assessing *PSEN1* methylation and expression and the development of amyloid plaques. We found that short-term perinatal supplementation was as effective as the longer post-weaning supplementation in repressing *PSEN1* expression and amyloid deposition in adult mice. These results highlight the importance of epigenetic memory and methyl donor availability during early life to promote healthy aging and stress the functional role of non-CpG methylation.

## 1. Introduction

Sporadic Late-Onset Alzheimer’s disease (LOAD) is the prevalent form of AD, but unlike for the Early-Onset (EOAD) genetic forms, the mechanisms leading to the development of this pathology are still unknown [1]. Several genetic traits have been described in LOAD; however, except for the ApoE4 allele, these polymorphisms show a low degree of association with the disease [2]. On the other hand, the association of AD with epigenetic traits seems to acquire more evidence and relevance, and fits particularly well with the observation that LOAD appears to be a multifactorial pathology with many associated modifiable risk factors [3,4]. As a matter of fact, many of these factors associated with LOAD can be categorized as “environmental” risk factors. It is currently widely accepted that environmental factors such as diet, pollutants, chemical species, and physical and psychological stressors can modify the epigenome or, under a slightly different point of view, the epigenetic mechanisms are “mediators” of environmental stimuli [5]. Environmental epigenetics describes how different stimuli can induce epigenetic modifications that have the potential to impact the physiological status in the short time or, if the effect is subacute, could drive a slow shift from normal to diseased aging [6]. Among these factors, specific micronutrients have the highest potential to impact epigenetic modifications, particularly DNA methylation, as they are directly involved in the biochemical reactions regulating methylation homeostasis [7]. “One-carbon metabolism” is a biochemical cycle in which S-adenosylmethionine (SAM), the methyl-donor in all the “transmethylation” reactions including the DNA methylation, is produced by the condensation of dietary methionine and ATP. After transferring its -CH_3_ group (methyl group) to DNA (or other substrates such as RNA, proteins, and lipids), SAM is transformed to S-adenosylhomocysteine (SAH), which is a feedback inhibitor of methyltransferases. Inhibition of methyltransferases is prevented by the rapid hydrolysis of SAH to adenosine and homocysteine (HCY), in the only reversible reaction of the pathway. HCY is secreted or transformed through the transsulfuration pathway (producing glutathione) and through the “remethylation” reaction leading to the reforming of methionine [8,9]. B vitamins, as enzymatic cofactors, are required for the reactions of both the transsulfuration (B6) and remethylation (B12 and folate/B9) pathways. A deficiency in these B vitamins is therefore potentially responsible for reduced HCY transformation and consequent accumulation. High HCY levels result in decreased SAH hydrolysis (since it is a reversible reaction) and consequent inhibition of DNA methyltransferases [10]. 

As a matter of fact, B vitamin deficiency and Methylation Potential (MP, defined as the SAM/SAH ratio) impairment has been detected in aging and AD subjects since the late 1990s [11,12,13,14]. These observations associating high levels of HCY, low levels of B vitamins and low MP with AD risk constituted the basis for our previous work, which demonstrated that B vitamin deficiency can be mechanistically associated with AD onset and progression. Indeed, we demonstrated that the methylation impairment caused by B vitamin deficiency resulted in the promoter hypo-methylation of *PSEN1* (Presenilin 1), a gene encoding for the peptide constituting the active site of the multiprotein 𝛾-secretase complex. This alteration leads to *PSEN1* over-expression, enhanced amyloidogenesis and increased senile plaque burden, which lead to worse behavioral scores in three-month-old TgCRND8 AD transgenic mice, which bear a doubly mutated human APP transgene and produce amyloid plaques starting at 2 months of age [15,16]. On the other hand, TgCRND8 mice supplemented with SAM from weaning until the age of three months showed a reduction in the AD-like phenotype induced by B vitamin deficiency [17]. The nutritional treatments also modulated other AD-associated factors, such as BACE1 (beta-secretase which processes APP at the beginning of the amyloidogenic pathway) but with mechanisms that are not mechanistically associated with DNA methylation [15,16,17].

Many studies have pointed out that epigenetic modifications in early life could be responsible for alterations in gene expression and, eventually, the onset of pathologies not only immediately but also in later life [18,19,20]. The principle at the basis of these theories is that the epigenetic changes are received and recorded during early life (perinatal), which is particularly prone to epigenetic changes, remembered throughout the many cellular divisions during the development and adult life of an organism and finally revealed in the late stages of life when the genes that have accumulated the epigenetic changes become critical for physiological aging processes [21].

Specifically, it has been largely proven that the availability of methyl donors during the perinatal period can alter the manifestation of phenotypes in the adult mice. The pioneering experiment on the agouti mouse model in which pregnant females were supplemented with methylation inhibitors or methyl donors, showed for the first time that epigenetic modifiers administered to the mothers could influence offspring phenotypes [22]. Therefore, although the transgenerational transmissibility of an acquired methylation pattern remains a possibility in mammals [23,24], the maternal transmission of epigenetic modifications is an accepted mechanism to influence the offspring phenotype, immediately and/or in adulthood and aging. This evidence induced us to explore the effects of one-carbon metabolism alterations during early life. Therefore, based on our previous results on SAM supplementation to adult TgCRND8 mice, and within the framework of a larger project aimed at exploring the role of *PSEN1* methylation and expression during neurodevelopment and neurodegeneration [25], we decided to investigate whether perinatal SAM supplementation could affect *PSEN1* expression and AD-like phenotypes in TgCRND8 mice. To this end, female mice were fed with SAM-supplemented diet from the time of mating to weaning and *PSEN1* methylation and expression and amyloid plaques deposition were measured in the offspring. These data were then compared with mice receiving SAM either prenatally and as adults (i.e., to mums from the time of mating to weaning and to pups after weaning) or only as adults (i.e., after the weaning).

## 2. Results

### 2.1. Experimental Set-Up

To assess the efficacy of the SAM supplementation during the perinatal period, female mice were supplemented with SAM from the time of mating to weaning and post-weaning. The offsprings’ *PSEN1* promoter methylation and expression levels were measured at different stages: in the embryos at embryonic day 14.5 (ED14.5), in the newborn mice at post-natal day 21 (PND21) and in adult mice at 3 months of age. Amyloid deposition was assessed in 3-month-old mice, which is the age that this strain shows evident senile plaques. The results were compared with mice who were fed the control diet and mice supplemented with SAM only after weaning, or both, during the perinatal period and after weaning (Figure 1a).

### 2.2. Exogenous SAM Uptake in Pups Brains

We previously demonstrated that orally supplemented SAM can be found in the brain of treated mice and, specifically, that the SAM-derived -CH_3_ group can be found in the brain DNA of these mice [17]. Our aim here was to determine whether the SAM supplemented to the mother could be found in the brain of the pups. After giving tritiated SAM by oral gavage to WT pregnant females or weaned mice, we analyzed the presence of radioactive molecules in the whole brain lysates of ED14.5 embryos, PND21 pups and 3-month-old mice. We observed that although adult mice directly treated with tritiated SAM showed higher levels of [^3^H], the embryos and pups of mothers treated with tritiated SAM also showed significant [^3^H] amounts in their brain lysates (Figure 1b). These results indicate that SAM orally supplemented to the mothers can pass to pups during pregnancy and/or lactation. 

### 2.3. PSEN1 Methylation in Mouse Brains

We assessed the methylation pattern of the *PSEN1* 5′-flanking region at the single-cytosine level through bisulfite modification followed by cloning of the PCR products and Sanger sequencing. The use of unbiased non-CpG methylation (MIPs) PCR primers allow us to assess both CpG and non-CpG methylation levels [25,26]. We assessed the *PSEN1* methylation pattern of DNA isolated from the prefrontal cortex and hippocampus of TgCRND8 mice at the different ages indicated in Figure 1a. The DNA methylation analysis was also performed on WT littermates, resulting in methylation profiles completely comparable with those observed in TgCRND8 mice (Appendix A), as expected on the basis of a previous analysis [25].

The histograms on the left side of Figure 2, Figure 3 and Figure 4 report the percent methylation of each CpG and non-CpG (CpA, CpT, CpC) cytosines in the 5′-flanking region (from cytosine 1212–1879, reported in the *x*-axis) of the *PSEN1* gene in cortical (light grey columns) and hippocampal (dark grey columns) brain tissues from TgCRND8 mice. In all the investigated conditions, the cortical and hippocampal methylation patterns were almost identical (*p* > 0.5). The 11 CpG sites present in this region are identified by a dot over the relative columns. The histograms on the right side of Figure 2, Figure 3 and Figure 4 report the average percent methylation of three groups of cytosines: all cytosines, CpG cytosines and non-CpG cytosines. Also, in this case, the overall methylation levels in the cortical and hippocampal tissues are almost identical (*p* > 0.05). Figure 5 reports the overall methylation levels of the same three groups of cytosines over the total number of cytosines in the analyzed sequence.

The methylation patterns at ED14.5 (Figure 2) were highly comparable between WT (Figure 2a) and TgCRND8 (Figure 2b) embryos from mothers fed the control diet (contingency test, *p* > 0.05). CpG moieties had the highest methylation level, which was above 80% for all the sites except the two proximal to the TSS (cytosines 1813 and 1852). A few non-CpG cytosines showed around 60% methylation and others had methylation levels between 30 and 40%. Many non-CpG cytosines had methylation levels between 10 and 20% or lower. Overall, the CpG methylation level was about 80% and non-CpG was below 20% in both WT and untreated TgCRND8 embryos (Figure 2d,e, respectively). Interestingly, TgCRND8 embryos from mothers treated with perinatal SAM supplementation (PN, Figure 2c) showed significantly higher DNA methylation levels at the two CpG sites proximal to the TSS (>80%) and at almost all the non-CpG sites (contingency test, *p* < 0.01 vs. Control). This general increase was clearly shown when considering the overall methylation level (Figure 2f) that rose to 40% (from less than 20%) for all cytosines, to 90% (from 80%) for CpG cytosines and to 30% (from less than 20%) for the non-CpG cytosines. The highest relative increase in DNA methylation at ED14.5 can be ascribed to non-CpG moieties.

At weaning (PND21, Figure 3), TgCRND8 pups from mothers treated with the control diet showed an unchanged methylation profile with respect to the profile at ED14.5 for the control diet group (Figure 3a, contingency test, *p* > 0.05) whereas the pups from mothers perinatally treated with SAM (PN, Figure 3b) showed higher methylation levels both with respect to age-matched pups from mothers given the control diet (contingency test, *p* < 0.01) and to ED14.5 embryos with perinatal SAM supplementation (*p* < 0.05). Again, the relative increase in methylation can be ascribed mainly to non-CpG sites, as showed by their methylation level of 40% (from less than 20%), and only in part to CpG sites which increased their methylation to 95% (from 70%) (Figure 3d).

At 3 months of age, we had four different experimental groups: (i) mice from mothers treated with the control diet (Ctrl, Figure 4a,e); (ii) mice from mothers treated with the control diet and then receiving post-weaning SAM supplementation until they were sacrificed (PW, Figure 4b,f)—this was the same treatment as in [17]; (iii) mice from mothers receiving perinatal SAM and then receiving post-weaning SAM supplementation (PN + PW, Figure 4c,g); and (iv) mice from mothers receiving perinatal SAM supplementation and then receiving the control diet after weaning (PN, Figure 4d,h). All three SAM-treated groups (PW, PN + PW, PN) showed similar *PSEN1* methylation patterns, with a significantly higher methylation level with respect to the Ctrl group (Fisher’s test, *p* < 0.05 for PW, *p* < 0.01 for PN + PW and PN). All three groups also showed slightly but not significantly (Fisher’s test, *p* > 0.05) lower methylation levels with respect to the PN-supplemented PND21 pups. No differences were observed between mice on the control diet at 3 months and PND21. The overall methylation analysis (Figure 4e–h) also revealed slight but not significant (Fisher’s test, *p* > 0.05) non-CpG hypomethylation in the PN group (below 30%) with respect to the other two groups (above 30%). The two CpG sites proximal to the TSS in SAM-treated mice showed control-like methylation levels at 3 months. The methylation profile of PW SAM-treated mice was comparable to the data that was previously obtained under the same conditions [17].

The representation used in Figure 5, where the methylation of all-Cs, CpGs and non-CpGs is divided by the total cytosines number rather than the number of cytosine in each group, clearly shows that the main and most dynamic methylation activity was related to non-CpG moieties.

To summarize, perinatal SAM supplementation (PN group) showed the same effect on *PSEN1* methylation as the longer PW, and even PN + PW, treatments and the methylation pattern was also conserved after the shift to the control diet at PND21. Moreover, in terms of relative increases, SAM supplementation seemed to mainly affect the non-CpG moieties. Interestingly, a CpC-rich sequence (1475–1503), previously reported as hypermethylated in conditions associated with low *PSEN1* expression [25], appeared to be particularly susceptible to SAM-induced hypermethylation. Moreover, another non-CpG “cluster” (1620–1726) showed a specific methylation profile since it was markedly demethylated in TgCRND8 with respect to WT mice at ED14.5 on the control diet, and then it became methylated with PN treatment and also on the control diet after PND21.

### 2.4. PSEN1 Expression in Mouse Brains

*PSEN1* expression was investigated at the mRNA and protein levels by real-time PCR and Western blotting (Figure 6 and Figure 7, respectively).

Despite the increased methylation observed (Figure 2a), *PSEN1* mRNA expression at ED14.5 in PN embryos from perinatally SAM-treated mothers was comparable to that of the control (Figure 6a). On the other hand, in PND21 pups at weaning, perinatal SAM supplementation was associated with significantly (ANOVA, *p* < 0.01) decreased *PSEN1* mRNA expression (Figure 6b). A similar significant (ANOVA, *p* < 0.01) down-regulation with respect to control was evident in mice at 3 months of age, independent of the type of SAM supplementation received (i.e., PN, PW and PN + PW) (Figure 6c). Unexpectedly, a trend towards increased inhibition in PN mice was seen, although these differences failed to reach statistical significance.

For *PSEN1* protein (Figure 7), a down-regulation comparable with the change in mRNA levels was seen. Figure 7a,c, respectively, shows representative Western blot images for ED14.5 embryos and 3-month-old mice. The histograms in Figure 7b,d show the quantification of the Western blot results including replicates.

As observed for DNA methylation, there were also no evident differences in *PSEN1* mRNA and protein levels between the cortex and hippocampus.

### 2.5. Amyloid Plaque Deposition

The immunohistochemical analysis of amyloid plaques in the brain was performed on 3-month-old animals because TgCRND8 mice show extensive plaque deposition at this age. Four high-magnification images were taken in both the frontal cortex and hippocampus, the two regions most affected by the development of amyloid plaques, to estimate the plaque area (%) and number of plaques (Figure 8).

Representative micrographs on the left (Figure 8a–e) and right (Figure 8f–j) of Figure 8 illustrate, respectively, the prefrontal cortex and hippocampal areas (scale bar = 100 μm) of WT mice (Figure 8a,f), where we predict the absence of amyloid plaques, control TgCRND8 mice (Figure 8b,g) and SAM-treated mice: PW (Figure 8c,h), PN + PW (Figure 8d,i) and PN (Figure 8e,j). Consistent with the characteristics of this mouse strain, large amyloid plaque deposition was detectable at 3 months of age (Figure 8b,g) and, confirming our previous results [17], PW SAM treatment largely reduced the plaque burden below the control levels (Figure 8c,h). PN + PW SAM supplementation had a comparable effect in reducing amyloid plaques with respect to the PW only treatment (Figure 8d,i). Interestingly, when the SAM supplementation was limited to the perinatal period (PN, Figure 8e,j), the reduction in plaque burden at 3 months of age was maintained similarly to that of the longer (PN + PW) and late (PW) treatments.

This result was quantified and showed that the diet supplemented with SAM significantly decreased the % area covered by amyloid plaques (Student’s test, PW and PN + PW, *p* < 0.001; PN *p* < 0.01; Figure 8k) and the number of plaques (Student’s test, *p* < 0.0001; Figure 8l) in the frontal cortex and hippocampus. The results are expressed as fold changes vs. the control diet condition (set equal to 1); the plaque area was calculated as the percent brain area occupied by plaques with respect to the total area analyzed.

## 3. Discussion

Our data demonstrate that perinatal SAM supplementation (i.e., SAM supplemented to the mothers, from mating to weaning, without the pups having been directly supplemented with the methyl donor) reduces the expression of *PSEN1* via DNA methylation of the 5′-flanking region of the gene, and also reduces the amyloid plaque burden in adulthood.

The results presented in this paper provide fundamental insights at three levels: first, we advanced the knowledge of the epigenetic modulation associated with AD and the perspective of epigenetic treatments for this disease; second, we reinforced the concept that early-life epigenetic modifiers affect aging; and third, we made the astounding observation that epigenetic modifications indirectly acquired (through the mothers) in early life are maintained in adults even in the absence of continued supplementation of the modifier. Moreover, these data strongly confirm that non-CpG methylation plays functional roles in regulating gene expression.

As a matter of fact, we expanded our previous results showing that *PSEN1*, involved in amyloidogenic processing, is modulated by the methylation status of its promoter region and that SAM supplementation can represent a possible intervention to modulate its expression without completely inhibiting its enzymatic activity, and thus mitigating senile plaque deposition and AD progression. We also demonstrated that early-life, time-limited (i.e., perinatal) SAM supplementation is as effective as prolonged supplementation to adult mice in modulating *PSEN1* and reducing senile plaques.

Our laboratory was the first (and we have continued to contribute data for the last 20 years) to show that *PSEN1* expression is modulated by the methylation status of its 5′-flanking region and that its modulation affects amyloid deposition and cognitive impairment in TgCRND8 mice [15,16,17]. Moreover, we and others demonstrated that the *PSEN1* promoter region is differentially methylated in LOAD [25,27,28]. We are aware that *PSEN1* is not the sole factor responsible for amyloid plaque deposition and in fact other genes have been found to be differentially methylated in AD patients or animal models. These include pro-inflammatory cytokines genes such as IL-1β and IL-6 [29,30], PICALM [31,32] and adiponectin [33]. Other genes have been found to be modulated in association with DNA methylation although not in a causative way, (e.g., BACE1) [17,34]. We can therefore hypothesize that AD-associated epigenetic changes are not limited to a single gene but rather they orchestrate a complex regulation of many genes, each one contributing in a multifactorial way to the onset and progression of the pathology [35,36,37,38,39]. The data presented here are therefore to be considered as a piece of a larger puzzle in which DNA methylation has the potential to coordinate a network of genes and factors involved in AD.

In addition to producing new and deeper knowledge of the risk factors and molecular mechanisms underlying neurodegenerative processes, the studies aimed at unravelling the role of DNA methylation in AD also have the potential to identify new biomarkers and therapeutic strategies [40,41,42]. In this framework, we have already demonstrated that *PSEN1* methylation changes in AD patients are detectable both in the brain and blood [25]. In terms of using epigenetic modulators as possible therapeutic agents in AD treatment, the data discussed here demonstrate that SAM could be considered as an oral supplement to treat the cognitive impairment associated with neurodegenerative processes [17,43].

The concept that early-life events, induced by environmental stimuli, may trigger neurodegenerative processes through epigenetic mechanisms is not new [44,45], but the demonstration of direct causal correlations was lacking. Here, we have demonstrated that SAM supplementation in the perinatal period is responsible for *PSEN1* promoter methylation at CpG and non-CpG sites in adult mice, thus reducing gene expression at the mRNA and protein levels and reducing amyloid deposition. It is noteworthy that short PN supplementation was as effective as prolonged PW supplementation and PN + PW treatment. This appears quite remarkable considering that the mice in the PN group never directly received SAM (it was supplemented to the mothers) and were shifted to the control diet at weaning, maintaining this diet until the 3-month endpoint. This means that SAM can pass through the maternal circulation to the embryo and then to the brain of the pups, as demonstrated by the measurement of the radioactively marked SAM in embryo brains. We previously demonstrated that SAM could pass the blood–brain barrier in adult mice [17]. A very interesting observation was that the DNA methylation pattern of the *PSEN1* promoter and the inhibition of *PSEN1* expression acquired during the perinatal period is maintained until the adulthood in absence of additional methyl donor supplementation. This means that an acquired “phenotype”, dependent on an acquired epigenetic signature, may be maintained even when the epigenetic mediator is removed. Although the brain is a post-mitotic tissue, it was not expected to maintain the methylation marks for months despite dynamic methylation and demethylation events. The data on the observed methylation changes induced by SAM supplementation may acquire even stronger relevance if we consider that methylation patterns and total methylation in mice treated with the control diet were remarkably stable over the ED14.5, PND21 and 3-month timepoints. The methylation profile in TgCRND8 mice was comparable to the profile observed in WT mice, as already indicated by previous studies in aging mice [25]. This observation reinforces the idea that DNA methylation is a physiological modulator of *PSEN1* expression and that this modulation is independent of the transgenic background since it was observed in both WT and TgCRND8 animals. With the idea of linking functional effects (i.e., the AD-like symptoms) to DNA methylation, we decided to limit the downstream analysis (*PSEN1* expression and plaque deposition) to the TgCRND8 mice, since the WT mice do not produce sufficient amyloid to be detected by immunohistochemistry.

The above-mentioned data can be better discussed by observing Figure 9, which is a graphical rendering of the relative variation in DNA methylation, and *PSEN1* mRNA and protein levels occurring with the three different treatments: PW (a), PN + PW (b) and PN (c). The methylation, mRNA and protein levels in control animals were used as the reference values and set equal to 1; only the line related to overall methylation is reported for convenience. The light grey areas on the left of each graph indicate the timing at which the values were not measured but “deduced” or “hypothesized” based on the first measured values (weaning for the PW group, ED14.5 for the other two groups). The dark grey boxes over the graphs indicate the timeframe of the SAM treatment. These graphs make it evident that while CpG methylation (red line) variation at the endpoint of 3 months was comparable in all the three groups, the non-CpG methylation variation was higher in the two groups receiving perinatal SAM supplementation. Specifically, non-CpG methylation was divided into the methylation levels of the CpC-rich sequence (1475–1503, dotted line) and the rest of the non-CpG moieties (green line). Also, the PN + PW group showed a final methylation variation similar to that of the PN group, despite the latter group being given SAM supplementation for a limited time in early life. Therefore, SAM treatment only affects CpG methylation to a certain extent and independently of the duration and timing of the treatment. On the other hand, non-CpG methylation had an almost 3-fold increase, and the CpC-rich stretch showed a 2-fold increase, at weaning in the PN and PN + PW groups, which then diminish slightly, compared to control. This evidence strongly supports the idea that non-CpG methylation has a functional role in modulating gene expression, as already reported by our [25,30,46,47,48] and many other studies [49,50,51]. Noteworthy, much of the evidence reporting unexpectedly high, differential, functional and dynamic non-CpG methylation were obtained from nervous tissues [52,53,54,55]. This may be not surprising since the nervous system has quite limited cell renewal abilities and that CpG methylation is maintained by DNA methyltransferase 1 (DNMT1) activity, which is dependent on the cell cycle, whereas non-CpG methylation is due to de novo DNMT3a and DNMT3b which still function in the absence of cell replication [56]. It is also interesting to note that the CpC-rich stretch at bases 1475–1503, previously identified as prone to methylation changes [25], showed methylation dynamics comparable to the overall non-CpG methylation changes (Figure 9, dotted line). A similar CpC-rich sequence was identified in the human *PSEN1* 5′-flanking region that shows differential methylation in control and AD patients [25]. This behavior supports this kind of CpC-rich stretch as the ideal “minimum” sequence to be analyzed to simplify the methylation assay for the possible application of *PSEN1* methylation as a novel biomarker. The second CpC-rich region (1620–1726), although showing a clear difference between TgCRND8 and WT mice at ED14.5, seems less interesting as a potential biomarker since it was not differentially methylated at PND21 and 3 months and since it is a longer region (100 bp) with fewer non-CpG cytosines and also one CpG site which was always heavily methylated.

Undoubtedly, these findings open the door to a plethora of further questions. First of all, we should explore whether other genes are similarly modulated by perinatal SAM supplementation. To date, previous data have shown that a very limited number of genes (6%) were modulated in PW SAM-treated mice [57]. We previously discussed this unexpected finding with the idea that SAM supplementation modulates methylation through the endogenous methylation pathway, the one-carbon metabolism, which can drive the targeting only to specific genes depending on the particular cellular and temporal context. This is a relevant difference between metabolic modulation of one-carbon metabolism and structural intervention on the DNA sequence by epigenetic drugs. However, it would be worthwhile to check if SAM supplementation in early life can have higher or even detrimental effects through the modulation of other genes. Moreover, it is a well-established concept that methyl donor availability is essential during preconception and early life to guarantee the proper and healthy development of the embryo [58,59,60].

The present study is therefore limited to the analysis of *PSEN1*; however, in the future, we aim to assess other factors that we previously found to be modulated by SAM in these mice, such as the BACE1 gene (involved in amyloid processing) [15,17] and PP2A phosphatase activity (involved in tau phosphorylation) [47]. The analysis of these and other factors involved in neurodegeneration could result in a comprehensive picture of the effects exerted by methyl donor supplementation in AD.

A second limitation of the present study is that, compared to our previous study in adult TgCRND8 mice supplemented with SAM after weaning [17], we could not include groups with B vitamin deficiency, which we previously used to simulate an AD-associated risk factor and a way to induce hypomethylation through the alteration of one-carbon metabolism. Unfortunately, due to the great importance of the B vitamins (folate, B12, B6) for proper embryonic development, mothers kept on B vitamin-deficient diets showed very low mating and birth rate, probably due to reduced fertility and major malformation or abnormal embryonic development.

On the basis of the observed “maintenance” of the methylation pattern when SAM supplementation was suspended, one could wonder whether the acquired and established pattern could also be transmitted to the offspring in a further generation, although the evidence so far indicates that trans-generational inheritance of DNA methylation is inefficient in mammals and particularly in humans [61]. According to the definition of “epigenetic memory” [62], the observed maintenance in adult animals of the methylation profile acquired during the perinatal period can be classified as “cellular memory” and “transcriptional memory”, i.e., the mitotically heritable epigenetic marks and transcriptional states established in the developmental period on the basis of environmental stimuli. It would be interesting, in the future, to check whether these changes can be also meiotically heritable in further generations.

## 4. Materials and Methods

### 4.1. Mice and Diets

TgCRND8 mice (TgCRND8 × 129Sv), carrying the double mutant form APP_695_ (KM670/671NL + V717F) were obtained from David Westaway (University of Toronto, Toronto, ON, Canada) and maintained in a heterozygous condition (TgCRND8^+/−^) by mating male TgCRND8 with female non-Tg (TgCRND8^−/−^) 129Sv mice (Charles River Laboratories). All animals were housed in an air-conditioned room (temperature 21 ± 1 °C, relative humidity 60 ± 10%) with a 12:12 h light/dark cycle (lights on from 8 AM to 8 PM) and food and water were continuously available. SAM (S-adenosylmethionine disulfate p-toluensulfonate) was obtained from “Gnosis by Lesaffre” (Desio, Milan, Italy) and was premixed with the diet pellets (Mucedola s.r.l., Settimo Milanese, Milan, Italy) at a concentration of 0.1 g/kg to obtain a SAM dosage of 400 μg/day as previously described [16,17].

WT females were mated with TgCRND8 males and, after checking for a plug, were randomly assigned to receive the control (Ctrl, N = 20) or SAM-supplemented diet (N = 20). The experimental set-up is depicted in Figure 1a: Group #1 (Ctrl) received the control diet from mating (moms) until adult age (pups/adults); Group #2 (PW, Post-Weaning) received the control diet from mating until weaning (moms) and SAM-supplemented diet after weaning (pups/adults); Group #3 (PN + PW, PeriNatal + Post-Weaning) received the SAM-supplemented diet from mating (moms) until adult age (pups/adults); and Group #4 (PN, PeriNatal) received the SAM-supplemented diet from mating until weaning (moms) and control diet after weaning (pups/adults). 

The brain tissues were collected as follows:-5 Ctrl mothers and 5 SAM-supplemented mothers were sacrificed at ED14.5 to obtain 10 TgCRND8 embryos for each diet arm (Ctrl and SAM-supplemented, Groups #1 and #2);-5 TgCRND8 pups for each diet group were sacrificed at PND21 (weaning) in order to collect 10 Ctrl (Groups #1 and #2) and 10 SAM-supplemented (Groups #3 and #4) pups;-10 TgCRND8 animals for each diet group were sacrificed at 3 months of age;-5 WT ED14.5 embryos, PND21 pups and 3-month-old mice on the Ctrl diet were also sacrificed to check the basal levels of the molecular parameters analyzed in this paper. Only the methylation data at ED14.5 are shown for WT mice (Figure 2A).

Genotype was assessed as previously described [17] on tail fragments of embryos, pups and adult mice.

Brain tissues were collected as previously described [15] and homogenized using a mechanical tissue homogenizer (TissueLyser II, Qiagen, Milan, Italy) in the appropriate buffers for metabolite analyses and DNA, RNA or protein extraction. Male and female mice were assessed in equal numbers and no differences between the two sexes were observed in any assay.

For the SAM uptake assays, [methyl ^3^H]-labeled SAM (Merck, Milan, Italy) diluted in water (100 μL final volume) was administered by oral gavage to two 129Sv WT females, as previously described [17], at a dose of 400 μg in a single administration at ED12.5 and PND19. After two days, the mice (the mother/embryos or the pups) were sacrificed. The brains were then dissected, separated in the two hemispheres and homogenized.

For immunohistochemistry assays, 3-month-old animals were sacrificed by perfusion with 4% paraformaldehyde (PFA) in 0.1 M PBS after deep anesthesia with an intraperitoneal injection of 50 mg/kg tiletamine/zolazepam (Zoletil; Virbac, Milan, Italy) and 10 mg/kg xylazine (Rompum; Bayer, Milan, Italy).

All procedures were carried out in accordance with the European Communities Council Directive (86/609/EEC) and were formally approved by the Italian Ministry of Health.

### 4.2. SAM Uptake Assay

For the SAM uptake assays, tissue homogenates from 129Sv mice treated with tritiated SAM were processed as previously described [17] to obtain whole tissue lysates. The samples were then mixed with scintillation solution and counted using a Beckmann 6500 LS counter; the results were normalized as cpm/μg of tissue.

### 4.3. DNA Methylation Profiling of CpG and Non-CpG Moieties

Assessment of CpG and non-CpG DNA methylation was performed by bisulfite DNA modification and genomic sequencing using the MMPS1BisF1 (forward) and MMPS1BisR1 (reverse) non-CpG Methylation-Insensitive Primers (MIPs) that were previously described in [16,26,46,63]. These primers allow for the unbiased amplification of *PSEN1* DNA sequences regardless of non-CpG methylation status. Briefly, DNA was extracted from brain tissues using the DNeasy Blood and Tissue Kit (Qiagen, cat. 69504) and a Qiacube. An EpiTect Bisulphite kit was used for bisulfite treatment and the PCR products were cloned using a PCR Plus Cloning Kit (Qiagen, cat. 59104 and 231224, respectively). At least 12 clones from 10 mice per group (except 5 mice for WT, as previously described) were analyzed per experimental condition using M13 primers for sequencing. The sequencing reactions were performed by the cycle sequencing method using an ABI PRISM 3130xl genetic analyzer (Thermo Fisher, Monza, Italy). The methylation status of any single cytosine in each sequenced clone was annotated. For each experimental sample, methylation % of every single cytosine was calculated as the number of methylated cytosines divided by the number of sequenced clones × 100 [46]. Average methylation was calculated as the total number of all-C, CpG and non-CpG methylation sites over the number of all-C, CpG and non-CpG cytosine moieties. Overall methylation was calculated as the total number of all-C, CpG and non-CpG methylation sites over the total number of cytosine (all-C) moieties. The overall methylation values were used to draw Figure 9 where the overall methylation was reported as the variation versus the methylation levels with animals fed the control diet set equal to 1; the same criteria were used to calculate the relative variation in *PSEN1* mRNA and protein expression in the same figure.

The *PSEN1* GenBank accession number and expected products size of the MIPs used for bisulfite analysis were previously described [16]. The primers used allowed us to assess the methylation status of the plus (5′->3′) DNA strand. The graphical structure of the mouse *PSEN1* promoter, outlining the regulatory regions and primer binding sites, was previously described [16]. Several positive and negative controls, aimed at checking conversion efficiency, were performed as previously described [26,46,63]. 

### 4.4. Quantitative Real-Time PCR Analysis

RNA was extracted from homogenized brain tissue with an RNeasy Lipid Tissue mini kit (Qiagen, Milan, Italy); 1 μg of total RNA was used for cDNA synthesis, and 1 μg of total cDNA was used for each real-time reaction. The quantitative analyses were performed in triplicate for each sample as previously described [16,17]. Total cDNA amounts were standardized by normalization to the β-actin control and expressed as the fold increase over control samples (i.e., Ctrl diet). The expression level of *PSEN1* was also normalized to the reference genes GAPDH and 18S with similar results. The oligonucleotides used as primers in the PCR reactions were described in [15]. 

### 4.5. Protein Analyses

Brain homogenates were lysed in 50 mM TRIS-HCl pH 7.4, 150 mM NaCl, 1% Nonidet P-40, 0.1% SDS, 0.5% sodium-desossicholate, 1 mM EDTA, 1 mM sodium-ortovanadate, 5 mM sodium fluoride and Complete Mini protease inhibitor cocktail (Roche Applied Science) according to the manufacturer’s instructions. A total of 90 μg of the protein extracts were run on a 12% SDS-PAGE gel, then blotted onto nitro-cellulose (BIO-RAD, Hercules, CA, USA). Western blot signals were acquired and analyzed by a Fluor-S densitometer and the Quantity One 4.6 software (BIO-RAD, Milan, Italy). Optical densities (O.D.) from at least three different experiments were calculated for each sample and normalized with the corresponding β-actin signal O.D.; the O.D. ratios were then compared and expressed as the average fold increase with respect to the Control diet value. A β-actin antibody was also used to normalize signals, with results similar to a 14,3,3β antibody.

The primary and secondary antibodies were described in [15].

### 4.6. Immunohistochemistry

Mouse brains were quickly dissected, fixed, paraffin embedded, serially sectioned from mid-sagittal plane to parietal plane (slice thickness, 9 μm) and mounted as previously described [17]. Endogenous peroxidases were inactivated with 0.3% H_2_O_2_ for 15 min at room temperature. Incubation with anti-Aβ monoclonal primary antibody DE2B4 (Acris, Germany) was carried out at 4 °C overnight; this antibody specifically recognizes Aβ aggregated in senile plaques. Following several washes with PBS, the immunoreactivity of biotinylated secondary antibodies was visualized using an ABC Elite Kit (Vector, Burlingame, CA, USA) with DAB-nickel as the chromogen and images were acquired using the Nikon universal software NIS-Elements 4.2 (scale bar = 100 μm). Four adjacent microscopic fields encompassing entire cortical and hippocampal regions were acquired. Analysis of morphometric images, plaque number per fields and the relative area of plaques were calculated using the Nikon universal software NIS-Elements 4.2. The sum of the areas for each region was calculated in µm^2^. The Aβ plaque number per field was independently determined by two operators and the relative area of plaques was calculated using MetaMorph 5.5 software tools. The online sagittal Mouse Brain Atlas from the Tennessee Mouse Genome Consortium website was used for brain region identification.

### 4.7. Statistical Analysis

One-way or two-way ANOVA and Tukey’s post hoc test were used for mRNA and protein expression evaluation. The contingency test and Fisher exact test were used for DNA methylation analysis. Two-tailed Student’s test was used for the immunohistochemical analyses. All histograms show the mean value ± standard deviation. Asterisks in figures indicate statistically significant differences. All the statistical analyses were computed using the SPSS software 27.

## 5. Conclusions

In conclusion, the data presented here strongly support the concept that environmental epigenetic modifiers, such as diet, can effectively drive the aging phenotype when manipulated in early life. Maintaining the proper DNA methylation could be a useful intervention in the prevention or treatment of AD and related dementias. 

*PSEN1* methylation, which is similarly modified in the brain and blood of AD subjects, was previously demonstrated [25] to be a potential new AD biomarker, with particular regard to the non-CpG methylation pattern. We are currently studying specific cytosine clusters that seem representative of the methylation status of the entire *PSEN1* 5′-flanking region, which could be potentially used to develop rapid methylation assays using whole blood. *PSEN1* methylation may not be sufficient to impact AD onset and progression, but DNA methylation has been shown to modulate a network of genes that together have the potential to drive healthy aging. 

The data presented here contribute the “big picture” of the role of methylation in AD. The maintenance of DNA methylation patterns, with functional effects on gene expression and through cell generations in the absence of the modifier, is a cellular epigenetic memory that may have a significant impact on physiology and pathology beyond neurodevelopment and neurodegenerative disorders.

## Figures and Tables

**Figure 1 ijms-24-11675-f001:**
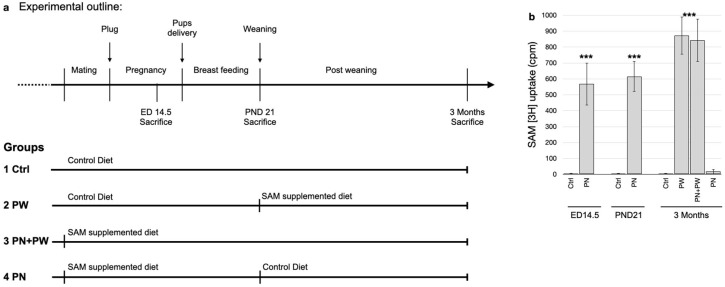
Experimental outline and SAM uptake. (**a**) Graphical schematic of the experimental set-up: the upper line shows the timing of the experiment and the different phases; the lower part shows the experimental groups and the diet administration. (**b**) [methyl-^3^H]-SAM uptake (cpm) was determined in total brain lysates in animals that received tritiated and non-tritiated SAM (*** *p* < 0.001, N = 5). Abbreviations: ED, embryonic day; PND, post-natal day; Ctrl, control diet/group; PW, post-weaning; PN, perinatal.

**Figure 2 ijms-24-11675-f002:**
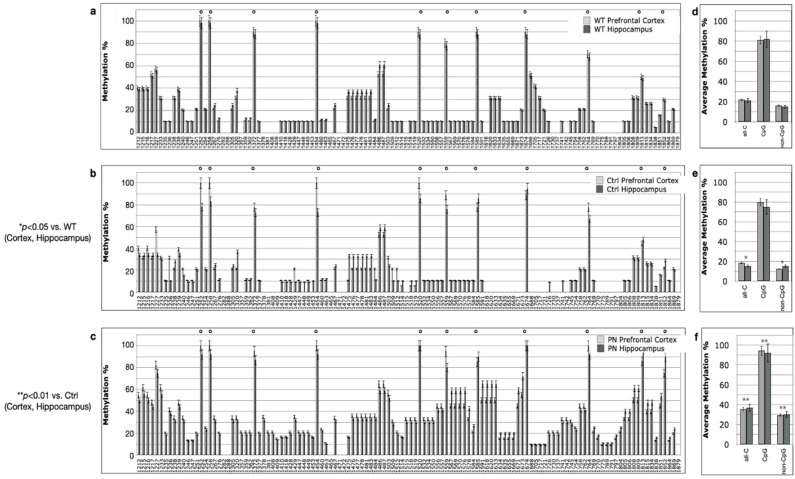
*PSEN1* promoter CpG and non-CpG methylation patterns in WT and TgCRND8 mouse brains at ED14.5. (**a,d**) WT, Ctrl diet, N = 5; (**b**,**e**) TgCRND8, Ctrl diet, N = 10; (**c**,**f**) TgCRND8, perinatal SAM-supplemented diet (PN), N = 10. Histograms in (**a**–**c**) (**left**) show methylation % (±standard deviation, *y*-axis) of each cytosine; labels on *x*-axis indicate the cytosine position on the reference sequence from the 5′ (**left**) to the 3′ (**right**) of the promoter, i.e., with the region proximal to the Transcription Start Site (**right**). CpG cytosines are marked by a dot over the related columns. Histograms in (**d**–**f**) (**right**) show the average methylation % over all cytosines (±standard deviation, *y*-axis) as derived from the respective data and grouped as all cytosines (all-C), CpG moieties (CpG) and non-CpG moieties (non-CpG). Light grey columns represent methylation in the prefrontal cortex, dark grey columns represent methylation in the hippocampus. The level of statistical significance using contingency table analysis is shown on the left side for TgCRND8 (**b**,**e**) vs. WT and TgCRND8 supplemented diet group (**c**,**f**) vs. Ctrl group.

**Figure 3 ijms-24-11675-f003:**
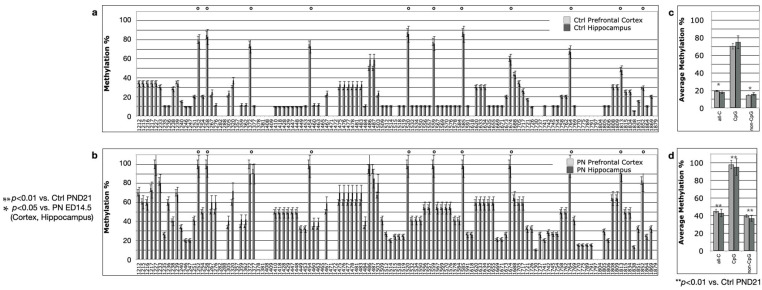
*PSEN1* promoter CpG and non-CpG methylation patterns in TgCRND8 mouse brains at PND21. (**a**,**c**) Ctrl diet; (**b**,**d**) perinatal SAM-supplemented diet (PN); N = 10 per group. Histograms in (**a**,**b**) (on the left) show methylation % (±standard deviation, *y*-axis) of each cytosine, organized as in Figure 2. CpG cytosines are marked by a dot over the related columns. Histograms in (**c**,**d**) (on the right) show the average methylation % over all cytosines (±standard deviation, *y*-axis) as derived from the respective data and grouped as all cytosines (all-C), CpG moieties (CpG) and non-CpG moieties (non-CpG). Light grey columns represent methylation in the prefrontal cortex, dark grey columns represent methylation in the hippocampus. The level of statistical significance using contingency table analysis is showed on the left of the figure for TgCRND8, perinatally supplemented diet group PND21 vs. Ctrl group and vs. TgCRND8, perinatally supplemented diet group ED14.5 on the left of (**b**,**d**).

**Figure 4 ijms-24-11675-f004:**
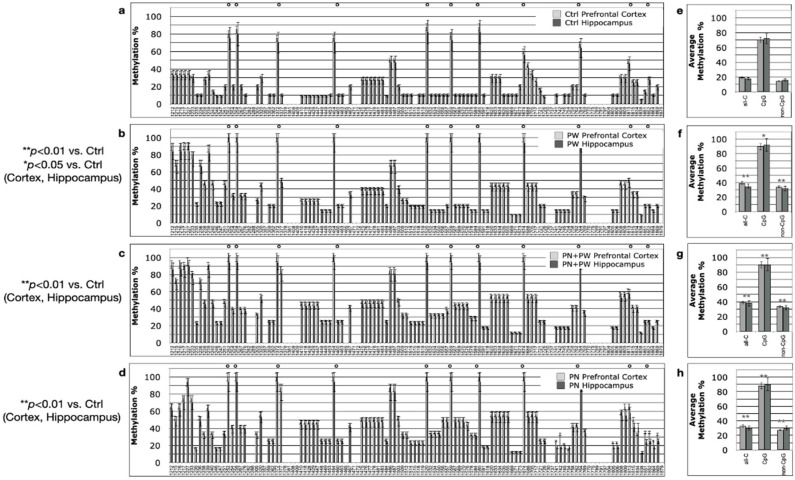
*PSEN1* promoter CpG and non-CpG methylation patterns in TgCRND8 mouse brains at 3 months. (**a**,**e**) Ctrl diet; (**b**,**f**) post-weaning SAM-supplemented diet (PW); (**c**,**g**) PW + perinatal SAM-supplemented diet (PW + PN); (**d**,**h**) perinatal SAM-supplemented diet (PN); N = 10 per group. Histograms in (**a**–**d**) (on the left) show methylation % (±standard deviation, *y*-axis) of each cytosine, organized as in Figure 2. CpG cytosines are marked by a dot over the related columns. Histograms in (**e**–**h**) (**right**) show the average methylation % over all cytosines (±standard deviation, *y*-axis) as derived from the respective data and grouped as all cytosines (all-C), CpG moieties (CpG) and non-CpG moieties (non-CpG). Light grey columns represent methylation in the prefrontal cortex, dark grey columns represent methylation in the hippocampus. The level of statistical significance using contingency table analysis is showed for each TgCRND8 supplemented diet group vs. Ctrl group on the left of (**b**–**h**).

**Figure 5 ijms-24-11675-f005:**
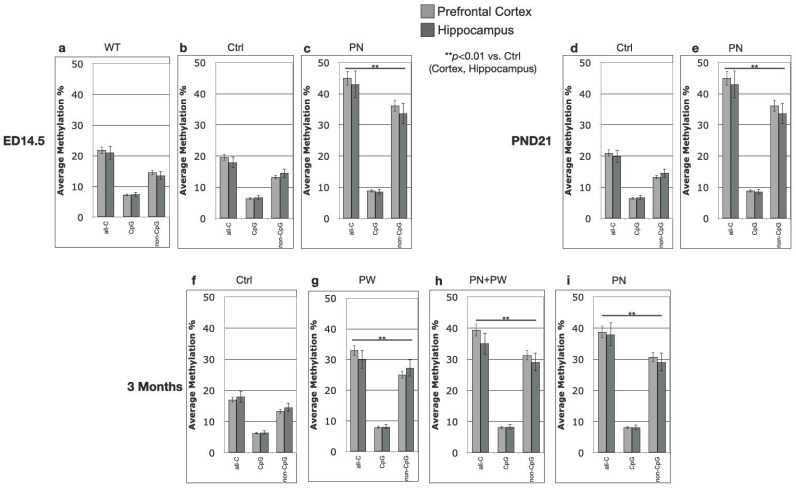
*PSEN1* promoter overall methylation in TgCRND8 mouse brains. Histograms show the overall methylation % (±standard deviation, *y*-axis) as derived from the respective data and grouped as all cytosines (all-C), CpG moieties (CpG) and non-CpG moieties (non-CpG) over the total number of cytosines moieties. (**a**–**c**), ED14.5; (**d**–**e**) PND21; (**f**–**i**) 3 months. Light grey columns represent methylation in the prefrontal cortex, dark grey columns represent methylation in the hippocampus. The level of statistical significance for all-Cs, CpGs and non-CpGs using contingency table analysis is showed under the histograms for each group vs. Ctrl group.

**Figure 6 ijms-24-11675-f006:**
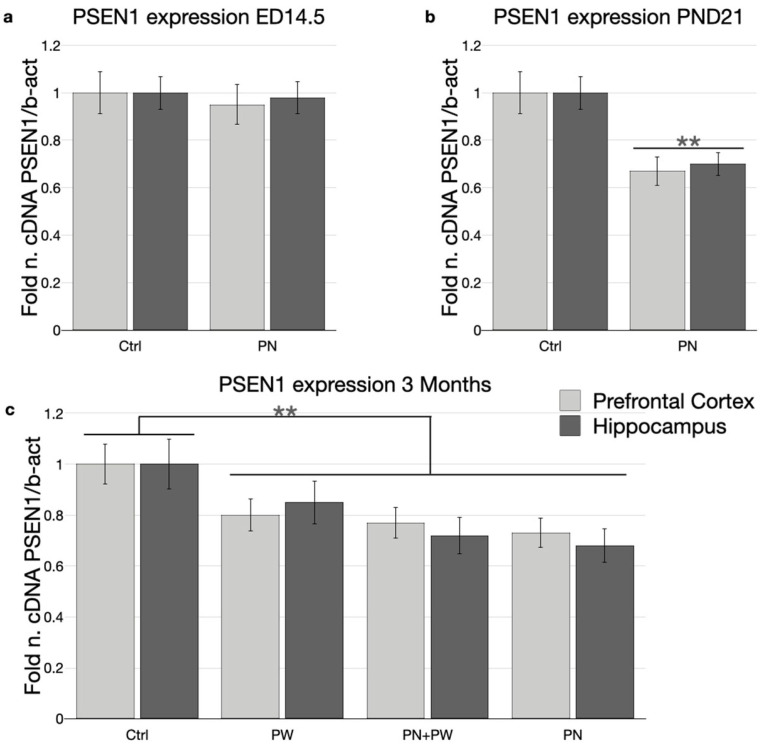
*PSEN1* mRNA expression in TgCRND8 mouse brains at ED14.5, PND21 and 3 months. (**a**) ED14.5; (**b**) PND21; (**c**) 3 months. Histograms show, on the *y*-axis, the relative amounts, obtained by real-time PCR, of *PSEN1* normalized to β-actin internal reference. Light grey columns represent mRNA expression in the prefrontal cortex, dark grey columns represent mRNA expression in the hippocampus. Abbreviations: ED, embryonic day; PND, post-natal day; Ctrl, control diet/group; PW, post-weaning; PN, perinatal. (** *p* < 0.01 vs. Ctrl, N = 10 per group).

**Figure 7 ijms-24-11675-f007:**
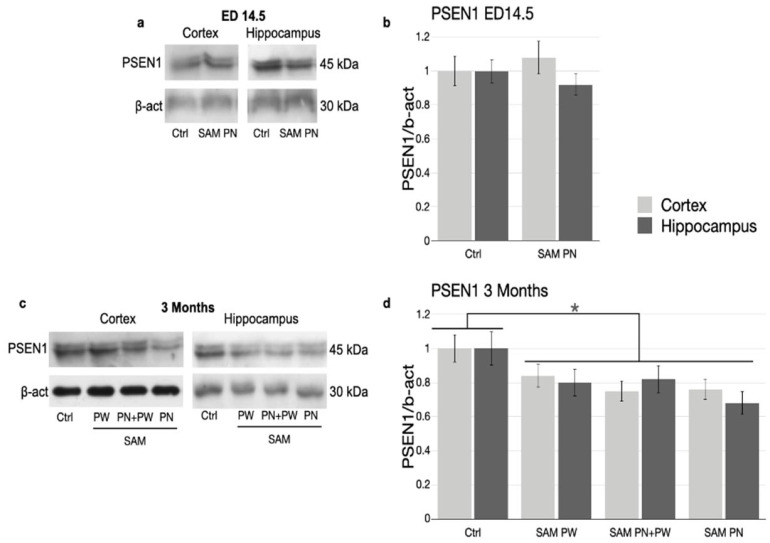
*PSEN1* protein expression in TgCRND8 mouse brains at ED14.5 and 3 months. Representative Western blot analyses of *PSEN1* and β-actin proteins were performed using brain lysates of TgCRND8 mice at (**a**) ED14.5 and (**c**) 3 months treated with Ctrl or SAM-supplemented diet. (**b**,**d**) Densitometric analysis of protein bands obtained in at least 3 independent experiments, using the β-actin band as internal reference. Light grey columns represent protein levels in the prefrontal cortex, dark grey columns represent protein levels in the hippocampus. Abbreviations: ED, embryonic day; Ctrl, control diet/group; PW, post-weaning; PN, perinatal. (* *p <* 0.05; N = 8 per group).

**Figure 8 ijms-24-11675-f008:**
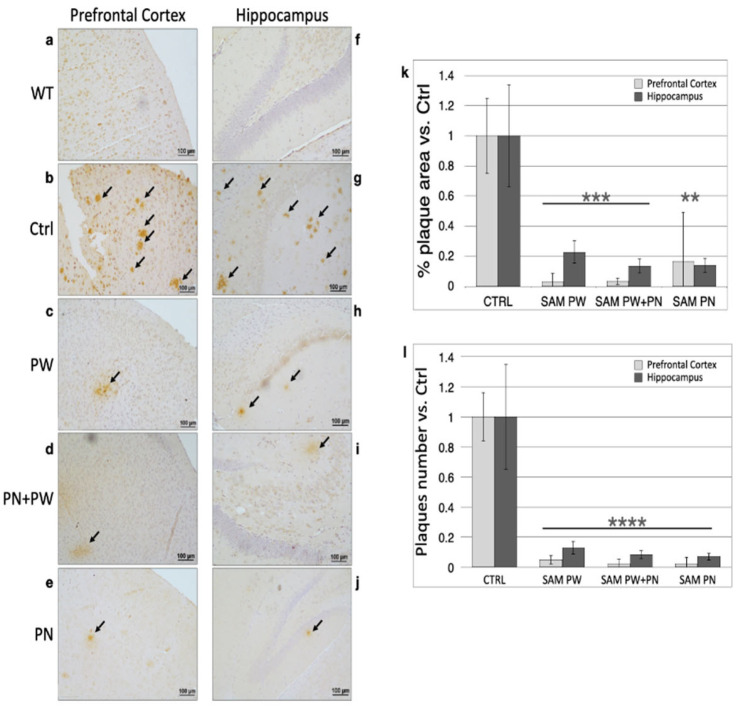
Amyloid plaque burden in TgCRND8 mouse brains at 3 months of age. From (**a**–**j**). Representative pictures of magnified 100 μm fields from the prefrontal cortex and hippocampus are shown. The left and right column pictures show, respectively, the prefrontal cortex and the hippocampus (bar = 100 μm) of (**a**,**f**) WT mice, (**b**,**g**) control TgCRND8 mice and SAM-treated TgCRND8 mice after the following treatments: (**c**,**h**) PW, (**d**,**i**) PN + PW, (**e**,**j**) PN. The histograms show the (**k**) % area covered by amyloid plaques and (**l**) plaque number in TgCRND8 mouse brains. Light grey columns represent amyloid plaques in the prefrontal cortex, dark grey columns represent amyloid plaques in the hippocampus. Abbreviations: Ctrl, control diet group; PW, post-weaning; PN, perinatal. (** *p* < 0.01, *** *p* < 0.001, **** *p* < 0.0001, vs. Ctrl; N = 8 per group except N = 3 for WT mice).

**Figure 9 ijms-24-11675-f009:**
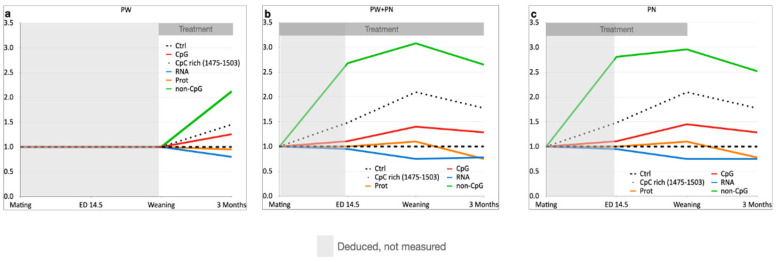
Schematic summary of *PSEN1* DNA methylation and expression at mRNA and protein levels in TgCRND8 mice. The three graphs schematically represent the levels of DNA methylation, mRNA expression (blue line) and protein expression (orange line). DNA methylation levels are divided into CpG methylation (red line), methylation at a CpC-rich sequence (bases 1475–1503, dotted line) and non-CpG methylation minus the CpC-rich sequence (green line). The black dashed line, set to 1, represents the reference level of overall DNA methylation, mRNA expression and protein expression; this line is the starting level observed in the control diet and against which the relative levels with SAM supplementation were calculated. The *x*-axis reports the timing of the analysis and the *y*-axis the relative value with respect to the control diet. The duration of the treatment with SAM is represented by the dark grey box over the graph. The light grey area on the left of each graph represents the interval in which the different outcomes were not measured but were deduced. (**a**) PW, (**b**) PN + PW and (**c**) PN SAM supplementation.

## Data Availability

The datasets used and/or analyzed during the current study are available from the corresponding author upon reasonable request.

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
