# Peer review of "Perinatal S-Adenosylmethionine Supplementation Represses PSEN1 Expression by the Cellular Epigenetic Memory of CpG and Non-CpG Methylation in Adult TgCRD8 Mice"

_ijms, 2023, doi:10.3390/ijms241411675_

Round 1
Reviewer 1 Report
This MS described the data here presented strongly supports the concept that environmental epigenetic modifiers, such as the diet, can be effective in early life and then drive the aging phenotype. Maintaining the proper DNA methylation could be a useful intervention in the prevention or treatment of AD and PSEN1 methylation could be considered for the possible development of a new AD biomarker, with particular regard to the non-CpG methylation. PSEN1 methylation per se could be not sufficient to impact AD onset and progression, but DNA methylation is shown to modulate a network of genes that together have the potential to drive healthy aging. Although this MS has overall interest and visibility, some aspects should still be considered to improve the quality and objectiveness.
1) The abstract is not clear. Please add the aim, objective and conclusion of the MS. Authors provided the results only.
2) The background of the study should be made very clear. Provide more details of the introduction and review of the work.
3) Please speculate about the reasons for the obtained results. The discussion needs to improve.
4) In Conclusion, the authors should add the potential practical application.
5) The article should be reviewed for English language proficiency and grammar. There are a lot of sentences without sense, misspelled words, and punctuation errors.
The article should be reviewed for English language proficiency and grammar. There are a lot of sentences without sense, misspelled words, and punctuation errors.
Author Response
We would like to thank the reviewers for their thoughtful and useful comments that helped us to improve the manuscript. Our point-by-point reply is presented below. Reviewers’ comments are in italic and the related replies follow.
Reviewer #1
.… Although this MS has overall interest and visibility, some aspects should still be considered to improve the quality and objectiveness.
> We are grate to the reviewer for the positive general comment.
1) The abstract is not clear. Please add the aim, objective and conclusion of the MS. Authors provided the results only.
> We are grateful for having raised this issue. We implemented the abstract according to the reviewer’s suggestion (in red in the MS). Please note that the journal’s format specifically indicates that the abstract is a single paragraph, ideally divided in background, aim, results and conclusion, but without subheadings.
2) The background of the study should be made very clear. Provide more details of the introduction and review of the work.
> We appreciated the suggestion of the reviewer. Since no specific indications about which part needed more detail, we tried to revise the whole introduction adding more specification in the parts we consider most difficult for people not directly involved inDNA methylation and AD. New and modified parte are in red in the MS.
3) Please speculate about the reasons for the obtained results. The discussion needs to improve.
> According to reviewer’s suggestion we tried to make more clear ho wand why the results are linked to the treatment. We also generally revised the discussion in the attempt to improve it. New and modified parte are in red in the MS.
4) In Conclusion, the authors should add the potential practical application.
> According to reviewer’s suggestion, we expanded the conclusion stressing the potential practical application (in red in the MS). We also moved the conclusions, previously included in the Discussion, to a dedicated section at the end of the MS (after the materials and methods) as allowed by the journal’s rules.
5) The article should be reviewed for English language proficiency and grammar. There are a lot of sentences without sense, misspelled words, and punctuation errors.
> The manuscript has been reviewed by the English language revision service offered by the publisher
Reviewer 2 Report
My recommendations for the research work Perinatal
S adenosylmethionine supplementation represses 2 PSEN1 expression by cellular epigenetic memory of CpG and 3 non CpG methylation in adult TgCRD8 mice are below;
My request to the authors is to follow it properly;
1. Ensure Spacing and punctuation errors throughout
2. Ensure all the abbreviations are defined during the first instance of usage
3. Following papers should be referred to by the author in the introduction; https://doi.org/10.1038/s41582-023-00789-z and https://doi.org/10.2174/1389200223666220310113110
4. Author should add a method and material section before the results
5. Study limitations should be defined clearly
6. A conclusion section will also add value
7. Some figures are not properly legible
Moderate editing of English language required
Author Response
We would like to thank the reviewers for their thoughtful and useful comments that helped us to improve the manuscript. Our point-by-point reply is presented below. Reviewers’ comments are in italic and the related replies follow.
Reviewer #2
Ensure Spacing and punctuation errors throughout
> We checked spacing and punctuation. The last has been also reviewed during the language correction
2. Ensure all the abbreviations are defined during the first instance of usage
> We checked the abbreviations and added definitions where necessary
3. Following papers should be referred to by the author in the introduction; https://doi.org/10.1038/s41582-023-00789-z and https://doi.org/10.2174/1389200223666220310113110
> We would like to thank the reviewer for having pointed our attention on the first paper. We used this reference to substitute an older one we indicated in the manuscript for the genetic basis of AD.
As for the second paper, the given link and DOI are referred to a paper related to ALS (Chavda VP, Patel C, Modh D, Ertas YN, Sonak SS, Munshi NK, Anand K, Soni A, Pande S. Therapeutic Approaches to Amyotrophic Lateral Sclerosis from the Lab to the Clinic. Curr Drug Metab. 2022;23(3):200-222) that is not related to any aspect of the present manuscript. We therefore respectfully decided not to follow this request.
4. Author should add a method and material section before the results
> According to the instructions and to the journal’s format, the Materials and Methods section is placed after the Discussion. We are afraid that, unless the journal allows a shift, we cannot accommodate this request.
5. Study limitations should be defined clearly
> The study limitations highlights, already present in the discussion, have been expanded and made clear (in red in the MS).
6. A conclusion section will also add value
> We are grateful to the reviews for the suggestion. The Conclusion section was previously included at the end of the Discussion section since not mandatory. We now expanded and moved the conclusions to a dedicated section at the end of the MS (after the materials and methods) as allowed by the journal’s rules.
7. Some figures are not properly legible
> We re-checked the figures submitted to be sure that all are correctly legible after uploading and that each one passed the journal’s QC
Moderate editing of English language required
> The manuscript has been reviewed by the English language revision service offered by the publisher